# Operationalising effective coverage measurement of facility based childbirth in Gombe State; a comparison of data sources

Josephine Exley[1]*, Antoinette Bhattacharya[1], Claudia Hanson[1,2], Abdulrahman Shuaibu[3], Nasir Umar[1], Tanya Marchant[1]

1 Department of Disease Control, London School of Hygiene & Tropical Medicine, London, United Kingdom, 2 Department of Public Health Sciences–Global Health, Karolinska Institutet, Stockholm, Sweden, 3 The Executive Secretary, Gombe State Primary Health Care Development Agency, Gombe, Nigeria

* Josephine.Exley@lshtm.ac.uk

**Data Availability Statement:** The comprehensive health facility project data used in this manuscript are freely available through LSHTM Data Compass

## Abstract

Estimating effective coverage of childbirth care requires linking population based data sources to health facility data. For effective coverage to gain widespread adoption there is a need to focus on the feasibility of constructing these measures using data typically available to decision makers in low resource settings. We estimated effective coverage of childbirth care in Gombe State, northeast Nigeria, using two different combinations of facility data sources and examined their strengths and limitations for decision makers. Effective coverage captures information on four steps: access, facility inputs, receipt of interventions and process quality. We linked data from the 2018 Nigerian Demographic and Health Survey (NDHS) to two sources of health facility data: (1) comprehensive health facility survey data generated by a research project; and (2) District Health Information Software 2 (DHIS2). For each combination of data sources, we examined which steps were feasible to calculate, the size of the drop in coverage between steps and the resulting estimate of effective coverage. Analysis included 822 women with a recent live birth, 30% of whom attended a facility for childbirth. Effective coverage was low: 2% based on the project data and less than 1% using the DHIS2. Linking project data with NDHS, it was feasible to measure all four steps; using DHIS2 it was possible to estimate three steps: no data was available to measure process quality. The provision of high quality care is suboptimal in this high mortality setting where access and facility readiness to provide care, crucial foundations to the provision of high quality of care, have not yet been met. This study demonstrates that partial effective coverage measures can be constructed from routine data combined with nationally representative surveys. Advocacy to include process of care indicators in facility summary reports could optimise this data source for decision making.

## Introduction

Ensuring access to high quality maternal and newborn care is a global priority in efforts to reduce preventable mortality and morbidity [1–5]. Measuring the quality of care delivered to

 

and can be accessed online: https://doi.org/10.17037/DATA.00001712.

**Funding:** This work is part of the IDEAS (Informed Decisions for Actions to improve maternal and newborn health) project. IDEAS is funded through a grant from the Bill & Melinda Gates Foundation to the London School of Hygiene & Tropical Medicine. Gates Global Health Grant Number: OPP1149259/INV-007644. TM is the PI of this grant. The funder had no role in study design, data collection and analysis, decision to publish or preparation of the manuscript.

**Competing interests:** I can confirm that the authors have declared that no competing interests exist.

women and newborns is central to supporting this goal, and effective coverage measures are now recommended as best practice [6–9]. Effective coverage combines need, use and quality of care into a single metric to estimate the proportion of a population in need of a service that had a positive health outcome from that service.

There is emerging consensus that effective coverage of maternal, newborn and child health (MNCH) is best conceptualised using a care cascade, which outlines six sequential steps that the target population is anticipated to have to move through to achieve the intended health benefit: 1) service contact; 2) input-adjusted coverage; 3) intervention-adjusted coverage; 4) process quality-adjusted coverage; 5) user-adherence adjusted coverage; and 6) outcome-adjusted coverage [7,10]. For health services, such as childbirth care, during which multiple interventions are delivered and the direct health impacts of specific interventions is challenging to attribute, process quality-adjusted coverage is the recommended measure of effective coverage.

Despite consensus on the concept, research on how to best operationalise the cascade, including which data sets and indicators to use, is limited [11–14]. A recent review of effective coverage of MNCH interventions found no consistent approach to the adjustments made to contact coverage, and examples in the literature were frequently limited to the use of primary sample survey data or use of open-access nationally representative surveys [15]; the review identified only one study that used data from health management information systems (HMIS) to estimate effective coverage of childbirth based on health outcomes [16]. To inform decisions at the country level there is a need to examine the extent to which meaningful effective coverage measures can be developed from alternative data sources that are routinely available to decision makers such as administrative or HMIS data [7].

This study aimed to address this issue by demonstrating whether and how the effective coverage of childbirth care can be generated using two different health facility data sources to adjust population-level data on contact coverage in Gombe State, northeast Nigeria. First, using comprehensive health facility survey data generated by a research project and second we examined the feasibility of replicating that measure using only routine data sources typically available to decisions makers.

## Materials and methods

### Ethics statement

Ethical approval for this study was obtained from the Nigerian National Health Research Ethics Committee (NHREC/01/01/2007), the State Ministry of Health Gombe State (ADM/S/658/Vol. II/66) and the London School of Hygiene & Tropical Medicine (22330). For the health facility project specific data, all potential participants were provided with a study information sheet and a consent form in English and Hausa. The in-charge of each facility gave written informed consent for the facility survey; written consent was also obtained from the birth attendant interviewed and women observed. Participation was voluntary and participants were free to withdraw at any time.

### Study setting

Gombe State is a predominantly rural (80%) and sparsely populated state in northeast Nigeria [17]. It is made up of 11 local government areas (LGAs) and 114 wards; about half of the population live in the State's central belt, made up of four LGAs.

The northeast region of Nigeria has some of the highest maternal and newborn death rates globally, estimated at 1,549 per 100,000 live births in 2015 and 33 per 1,000 live births in 2017, respectively [18,19]. Healthcare is predominantly delivered via a network of rural primary

healthcare clinics (PHCs) run by the Gombe State Primary Healthcare Development Agency (GSPHCDA). In 2017, 460 PHCs and 26 referral facilities provided intrapartum services [20], mainly delivered by community health extension workers (CHEWs), junior CHEWs and community health officers.

Between 2016 and 2019, the GSPHCDA led a maternal and newborn health partnership designed to implement a package of evidence-based interventions to improve access, use and quality of maternal and newborn health services, across the 11 LGAs of Gombe State [21–24]. Throughout this period, actions were taken to strengthen the use of HMIS data for decision making [25], plus detailed primary data was collected to track progress in access to, and supply of, quality maternal and newborn health services.

## Data sources

Generating effective coverage of childbirth requires linking care seeking data collected through population based data sources with information from health facilities on the quality of the interventions provided [11]. Two sources of population data representative at the national and State levels in Nigeria had potential for this analysis: the Nigerian Demographic and Health Survey (NDHS) last conducted in 2018 and the Multiple Indicator Cluster Survey (MICS) last carried out in 2016/2017. We used NDHS given it was undertaken most recently. Two sources of facility data were accessed: 1) comprehensive health facility survey data collected as part of the partnership to improve maternal and newborn health services [24]; and 2) HMIS data available through monthly facility reports from District Health Information Software 2 (DHIS2). Previous studies of effective coverage of childbirth have used service provision assessment (SPA) and service availability and readiness assessment (SARA) [12,26–31], but neither of these surveys have been undertaken in Nigeria.

**Population data.** The NDHS is conducted every five years using a two-stage stratified cluster sample, designed to be representative at the national and state level [32]. The household survey included face-to-face interviews with all women aged 15 to 49 years in the sampled households, both permanent residents and visitors who stayed in the household the night before the survey. Data was extracted from the birth record for all women in Gombe State aged 15 to 49 who reported a live birth and the place of care seeking in the five years preceding the survey.

**Project specific health facility data.** We used health facility survey data from August 2019. Data collection methods are reported in detail elsewhere [33]. Briefly, a health facility survey was completed in a sample of 98 PHCs across the 114 wards of Gombe State and all 18 referral facilities in the State. The health facility survey comprised a readiness assessment that included a checklist of staff, equipment, drugs, and infrastructure items present on the day of survey; data extraction from facility registers on the number and outcomes of all births during the previous six-months; interviews with birth attendants; and the observation of births in a sub-set of facilities. For the purpose of this analysis, facilities handling fewer than one delivery per week (n = 11) were excluded on the grounds that they are not representative of the typical facility women seek childbirth care from.

During the facility survey, observations of childbirth were completed in the 10 PHCs with the highest number of births recorded in the maternity register [34,35]. Observations were completed by clinically trained female data collectors (local midwives, not employed by the facility) over a three-week period, using a structured checklist to record the processes of care and birth attendant-client interactions. The content of the checklist was developed from the USAID-funded Maternal and Child Health Integrated Program's tool for observing vaginal birth [36].

**Routine health facility data.** DHIS2 is an open source software platform used in more than 70 countries [37]. In Gombe, health facilities document care in 13 paper-based registers. Every month a sub-set of data in these registers is sent to the LGA health office and entered into DHIS2 [20]. Monthly aggregated DHIS2 data related to maternal and newborn health were downloaded for the same 6-month period as the project data, from January to July 2019. As with the project data, facilities that recorded fewer than one delivery per week on average were excluded.

## Operationalising the effective coverage cascade

We computed both effective coverage measures based on the coverage cascade steps for facility based childbirth care proposed by the Effective Coverage Think Tank Group–a group of experts convened by WHO and UNICEF [7]. Consistent with that cascade, we defined effective coverage as the proportion of all women with a recent live birth (the target population) who progressed through the subsequent four steps: 1) attended a health facility for childbirth care (service contact coverage), 2) that had appropriate inputs available (input-adjusted coverage), 3) where appropriate interventions were provided (intervention-adjusted coverage), and 4) where birth attendants followed recommended processes of care (process quality-adjusted coverage).

Table 1 shows how the effective coverage cascade was operationalised in the two combinations of data sources. For both cascades, the 2018 NDHS was used to estimate service contact

**Table 1. Overview of measures used to define each step of the coverage cascade for the different data sources: (1) NDHS and project data and (2) NDHS and DHIS2.**

| Step of the coverage cascade | (1) NDHS and project data | | (2) NDHS and DHIS2 | |
|---|---|---|---|---|
| | **Measures** | **Data source** | **Measures** | **Data source** |
| **Service contact coverage** | Facility based delivery among women with a live birth in last 5 yrs | NDHS | Facility based delivery among women with a live birth in last 5 yrs | NDHS |
| **Input-adjusted coverage** | Infrastructure:<br>• Means of communicating with another facility<br>• Electricity or alternative power supply.<br>• Accessible toilet<br>• Clean water | Health facility survey | | DHIS2 |
| | Staffing:<br>• Midwife/clinician available 24/7 | | Staffing:<br>• Skilled birth attendant | |
| | Drugs & commodities:<br>• Anticonvulsant<br>• Baby scale<br>• Blood pressure machine<br>• Delivery pack<br>• Intravenous fluids with infusion set<br>• Infection control inside labour room<br>• Newborn resuscitation device<br>• Suction apparatus<br>• Uterotonic | | Drugs & commodities:<br>• Anticonvulsant<br>• Newborn resuscitation device<br>• Uterotonic | |
| **Intervention coverage** | • Baby weighed<br>• Prophylactic uterotonic<br>• Thermal care | Observations of care | • Baby weighed<br>• Prophylactic uterotonic<br>• Thermal care | NDHS |
| **Process quality-adjusted coverage** | • Explains procedure to woman or support person before proceeding<br>• Maternal blood pressure taken during first stage of labour<br>• Support person (companion) for mother present at birth<br>• Woman recommends someone else to give birth in the health facility | Observations of care | - | - |

coverage (step 1). To define the content of input-adjusted (step 2), intervention-adjusted (step 3) and process quality-adjusted coverage (step 4), we undertook a review of the literature to examine how effective coverage of childbirth has previously been defined [15]. The review identified little consistency between study definitions. We therefore selected the most frequently cited items from the literature that were also recommended by WHO [38–41]. Selected items were mapped against data available in the comprehensive project datasets and the final selection was agreed upon between the authors, including the Executive Secretary of GSPHCDA to ensure relevance to the setting. We attempted to replicate the cascade using only data typically available to decision makers; where information relating to care received was not available in DHIS2, data from NDHS was applied. No items were available in either DHIS2 or NDHS that allowed us to estimate process quality-adjusted coverage. See S1 Table for full details of the individual data items used to define each step of the coverage cascade for the two approaches.

Input-adjusted measures were estimated in the respective health facility dataset (project data or DHIS2) as a binary score calculated for each facility based on: 1) all items available and functioning on the day of the survey in the project data, and 2) not experiencing stock outs of any items in the previous six months in DHIS2. Mean input-adjusted score was calculated, by facility type (PHC or referral), as the percentage of facilities with inputs available. For the project data mean intervention-adjusted and process quality-adjusted measures were estimated in the observation dataset, as the percentage of women observed in a PHC who received all components of care. In the data typically available to decision makers, intervention-adjusted coverage was calculated based on women's self-reports in the NDHS, as the percentage of women who reported they gave birth in a facility that received all interventions. All items contributed equally to each score and missing data was treated as the item not being present.

## Analysis

Similar to previous examples, effective coverage was calculated at the State level using ecological linking methods [11–14,31,42]. For both analysis, the NDHS was used as the basis for creating each linked dataset. Each woman in the NDHS who reported attending a facility for childbirth was assigned the mean input-adjusted score for the type of health facility (PHC or referral) that they reported seeking care from the project data and DHIS2, respectively. Additionally, for the analysis using the project data women were assigned the mean intervention-adjusted and process-adjusted score from the project data. In both analysis, women who reported delivering at home were assigned input-adjusted, intervention-adjusted and process quality-adjusted scores of 0.

From the linked datasets, we calculated each step of the cascade. The first step in the cascade, service contact coverage, was calculated as the percentage of women who reported giving birth in a facility across the State. Subsequent steps were calculated as the product of the prevalence of the step and the prevalence of the proceeding step. The analyses adjusted for the survey design using the svyset and svy commands in STATA version 15.1 (StataCorp, 2017, College Station, TX) and uncertainty of the estimates of effective coverage was assessed using the delta method [14,43]. Missed opportunities (bottlenecks) were identified from the absolute attrition in the proportion between each step of the cascade [44].

## Results

The analysis included 822 women who reported a live birth in Gombe State in the five years preceding in the NDHS (2013–2018) (Table 2). The project data included 105 health facilities (87 PHCs and 18 referral), which recorded handling at least one delivery per week and

**Table 2. Overview of study population for each dataset.**

| Women interviewed | NDHS–Gombe State | |
|---|---|---|
| Number of women interviewed with a live birth in the last five years | 823 | |
| Number of women interviewed with a live birth in the last five years & place of birth recorded | 822 | |
| *Health facilities* | **Project data** | **DHIS2** |
| Number of PHCs | 98 | 547 |
| Number of PHCs with at least 1 delivery per week | 87 | 248 |
| Median number of births in PHCs in last 6 months (IQR) | 125 (64–192) | 66 (41.5–133.5) |
| Number of referral facilities | 18 | 26 |
| Number of referral facilities with at least 1 delivery per week | 18 | 23 |
| Median number of births in referral facilities in last 6 months (IQR) | 222.5 (154–573) | 239 (111–495) |
| Number of women observed during childbirth[1] | 398 | n/a |

NOTE: [1] Observations were completed in 10 PHCs.

observations of 398 women from 10 PHCs during childbirth. The analysis using data typically available included 271 health facilities (248 PHCs and 23 referral) from DHIS2.

Table 3 presents the characteristics of all women with a recent live birth interviewed in Gombe State in the NDHS. On average women interviewed were 29 years old (sd 4.7) and had received 3 years (sd 4.7) of education. The vast majority of women reported that they were currently married and of Muslim faith; fourteen percent reported they had one child.

In the rest of the results section we first describe the composition of the four steps of the cascade in turn and then present the two effective coverage measures estimated using the different data sources.

## Step 1: Service contact

In the NDHS for Gombe State, representing births between 2013–18, the coverage of facility based childbirth was 30%: 19% at PHCs and 11% at a referral facility. We checked for evidence of changes in facility delivery over the period of the NDHS, and found the coverage of women seeking childbirth care at a health facility to be relatively stable over the five-year period: 37% among women who delivered five years preceding the survey, 31% in the three to four years preceding, 32% in two years preceding and 27% among women who delivered in the 12 months preceding the survey.

## Step 2: Inputs

The availability of inputs from the two facility data sources (project health facility survey or DHIS2) by facility type is presented in Table 4. Around a quarter of facilities were estimated to have all inputs available in the project health facility data: 18% of PHCs had all inputs compared to 56% of referral facilities. Across all facilities communication equipment and disposable gloves was universally available. Additionally, among referral facilities electricity or light source, presence of a skilled birth attendant, blood pressure machine, delivery pack, infection control supplies, intravenous fluids and infusion set, suction apparatus and uterotonic were also universally available. The items least frequently available in PHCs were source of cleaning running water (56%) and presence of a skilled birth attendant (49%), and in referral facilities source of clean running water and newborn resuscitation equipment (both available in 78% of referral facilities).

**Table 3. Characteristics of women interviewed in Gombe State NDHS with a recent live birth and place of birth recorded, column percentage.**

| Characteristic | | % (95% CI) |
|---|---|---|
| Age | 15–19 | 6.3 (4.7–8.4) |
| | 20–29 | 45.6 (41.4–49.8) |
| | 30–39 | 37.9 (32.2–41.2) |
| | 40–49 | 11.2 (9.0–13.9) |
| Schooling | None | 72.1 (60.1–81.7) |
| | 1–7 years (primary) | 10.3 (6.9–15.2) |
| | ≥ 8 years (secondary) | 17.5 (10.4–28.0) |
| Religion | Christian | 14.0 (6.8–26.4) |
| | Muslim | 85.9 (73.4–93.1) |
| Parity | 1 birth | 14.1 (11.8–16.7) |
| | 2 births | 13.1 (10.3–16.5) |
| | 3–5 births | 34.4 (32.4–36.5) |
| | ≥ 6 births | 38.4 (34.4–42.6) |
| Marital status | Currently married | 94.8 (91.3–97.0) |
| | Not currently married | 5.2 (3.0–8.7) |

The number of input measures it was possible to estimate in the DHIS2 was limited; no information was available on facility infrastructure and data was only captured on three of the 10 supply and commodity items included in the project-based estimate. Less than a fifth of facilities had all inputs available: 18% of PHCs and 22% of referral facilities. No items were universally available. The item most frequently available in PHCs was skilled birth attendant (73%) and uterotonic in referral (83%), while the item least frequently available in PHCs was anticonvulsant (34%) and in referral facilities skilled birth attendant (35%).

## Step 3: Receipt of interventions

Over three-quarters of women were observed to receive all three interventions in the project data (see Table 5); ranging from 99% of women receiving a uterotonic to 87% of babies being weighed. Since the DHIS2 did not capture equivalent information, the second effective coverage measure took available data on receipt of interventions from the NDHS. In the NDHS 5% of women reported that they received all interventions; ranging from 75% of women receiving a uterotonic to 13% of babies being weighed.

## Step 4: Process of care

Process of care data was available in the project data but not the data typically available to decision makers (see Table 6). Overall, 24% of women were observed to receive all four processes

**Table 4. Facility input measures used in the summary variable that resulted in 'input-adjusted' coverage in the cascade.**

| | Project data health facility assessment | | | DHIS2 | | |
|---|---|---|---|---|---|---|
| | PHC | Referral | All | PHC | Referral | All |
| **Facility infrastructure** | | | | | | |
| Communication equipment | 100 | 100 | 100.0 | - | - | - |
| Electricity or light source | 96.6 (92.7–100) | 100 | 97.1 (94.0–100) | - | - | - |
| Source of clean running water | 56.3 (45.8–66.79) | 77.8 (58.5–97.1) | 60.0 (48.0–72.0) | - | - | - |
| Toilet accessible to female service users | 82.8 (74.8–90.7) | 94.4 (83.8–100) | 84.8 (76.3–93.2) | - | - | - |
| **Staffing** | | | | | | |
| Skilled birth attendant | 49.4 (38.9–60.0) | 100 | 58.1 (49.3–66.8) | 72.6 | 34.8 | 69.4 |
| **Supplies and commodities** | | | | | | |
| Anticonvulsants | 82.8 (74.8–90.7) | 83.3 (66.0–100) | 82.9 (73.3–92.4) | 34.3 | 65.2 | 36.9 |
| Baby weighing scale | 97.7 (94.5–100) | 94.4 (83.8–100) | 97.1 (92.7–100) | - | - | - |
| Blood pressure machine (sphygmomanometer) | 93.1 (87.8–98.5) | 100 | 94.3 (89.9–98.7) | - | - | - |
| Delivery pack | 85.1 (77.5–92.6) | 100 | 87.6 (81.4–93.9) | - | - | - |
| Disposable gloves | 100 | 100 | 100 | - | - | - |
| Infection control in service area | 88.5 (81.8–95.2) | 100 | 90.5 (84.9–96.1) | - | - | - |
| Intravenous fluids and infusion set | 93.1 (87.8–98.5) | 100 | 94.3 (89.9–98.7) | - | - | - |
| Newborn resuscitation device | 77.0 (68.1–85.9) | 77.8 (58.5–78.6) | 77.1 (66.5–87.8) | 41.1 | 73.9 | 43.9 |
| Suction apparatus | 93.1 (87.8–98.5) | 100 | 94.3 (89.9–98.7) | - | - | - |
| Uterotonic | 96.6 (92.7–100) | 100 | 97.1 (94.0–100) | 51.2 | 82.6 | 53.9 |
| **ALL INPUTS AVAILABLE** | 18.4 (10.2–26.6) | 55.6 (32.5–78.6) | 24.8 (14.0–35.5) | 17.7 | 21.7 | 18.1 |

**Table 5. Receipt of intervention measures used in the summary variable that resulted in 'intervention-adjusted' coverage in the cascade.**

| | Project observations of care | NDHS | | |
|---|---|---|---|---|
| | | PHC[1] | Referral[1] | All |
| **Interventions** | | | | |
| Baby weighed | 87.2 (73.2–100) | 7.1 (2.7–11.5) | 24.4 (13.2–35.5) | 13.4 (8.0–18.8) |
| Prophylactic uterotonic | 98.5 (97.1–99.9) | 75.9 (67.8–84.0) | 74.0 (66.1–81.9) | 75.2 (69.9–80.5) |
| Thermal care | 89.5 (81.8–97.1) | 64.1 (56.7–71.5) | 48.9 (36.9–60.9) | 58.5 (52.8–64.2) |
| **ALL INTERVENTIONS RECEIVED** | **78.4 (64.0–92.8)** | **4.6 (1.1–8.0)** | **6.0 (1.9–10.0)** | **5.1 (2.5–7.6)** |

NOTE:

[1] NDHS coverage data calculated separately for women reporting attending a PHC or a referral facility for childbirth care.

**Table 6. Process of care measures used in the summary variable that resulted in 'quality-adjusted' coverage in the cascade.**

| Process of care | Project observations of care |
|---|---|
| Takes woman's blood pressure | 49.8 (31.3–68.2) |
| Explains procedure to woman or support person before proceeding | 70.4 (63.4–77.3) |
| A support person (companion) for mother present at birth | 54.3 (31.5–77.0) |
| Mother would recommend someone else to deliver in the facility | 94.2 (88.5–99.9) |
| **ALL PROCESSES OF CARE RECEIVED** | **24.1 (9.9–38.3)** |

of care. Across the three items undertaken by the birth attendant coverage ranged from 50% observed to take women's blood pressure to 70% observed explaining a procedure.

### Effective coverage of facility based childbirth

Fig 1 presents the coverage cascade for facility based childbirth care in Gombe estimated using project data and datasets typically available to decision makers. NDHS was used in both estimates to estimate service contact (step 1). For the first effective coverage measure using the project data to calculate effective coverage from cascade steps 2 to 4, we observed that 2% of women in Gombe received high quality care during childbirth. The largest bottleneck was in access to a health facility; only 30% of women reported attending a health facility for childbirth. There was also a large reduction from service contact to input-adjusted coverage, with an attrition of 21%. The drop from input-adjusted coverage to intervention-adjusted coverage was relatively small, from 10% to 7%, reflecting the high percentage of women receiving all interventions.

For the second effective coverage measure using data typically available to decision makers in this setting we were able to calculate effective coverage up to cascade step 3. We observed

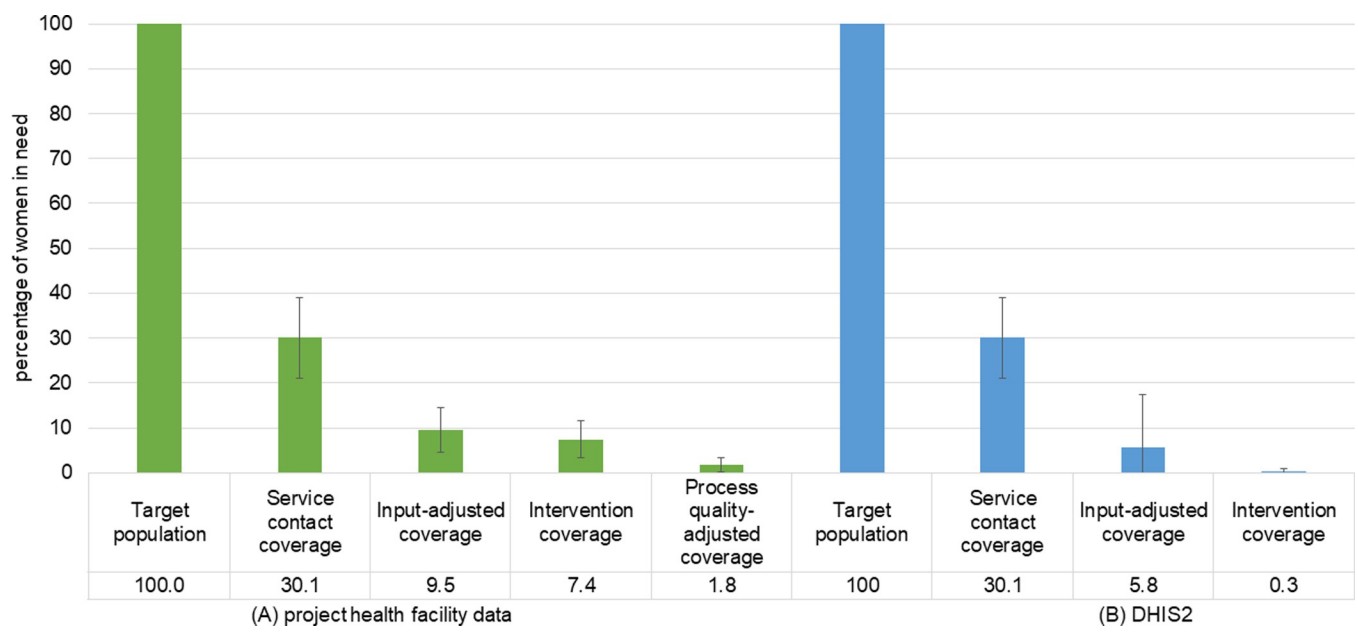

**Fig 1.** Effective coverage of facility based childbirth in Gombe State, constructed using (a) NDHS and project facility data; and (b) NDHS and DHIS2.

that less than 0.5% of women were estimated to receive high quality care during childbirth. Again the largest bottleneck was in access to a health facility with only 30% of women attending a health facility for childbirth, and there was a large reduction from service contact to input-adjusted coverage, 30% to 6%.

## Discussion

Effective coverage measures are recommended as best practice for estimating population-level access to high quality maternal and newborn health care but there has been limited progress to operationalise measures. To maximise utility, there are increasing calls to make better use of routine data systems to generate estimates of effective coverage [7]. In this study, using different health facility data sources to estimate the effective coverage of facility based childbirth, we aimed to determine the feasibility of using data typically available to decision makers in this high mortality setting.

Our first approach linking NDHS to health facility survey data collected through a research project represents the most comprehensive health facility data available in this setting. It included a health facility survey and observations of birth, allowing linking of these different data sources with NDHS to calculate all four recommended cascade steps to estimate process quality-adjusted coverage of childbirth. The analysis revealed that less than 2% of women received effective coverage of childbirth care in Gombe state. Substantial gaps in the provision of high quality care were highlighted; coverage dropped from 30% of women who attended a facility for childbirth to 10% after accounting for the necessary inputs to provide high quality care during childbirth, dropping again to 7% after adjusting for intervention delivery and 2% after finally adjusting for processes of care. This finding adds to the wealth of evidence demonstrating large drops in coverage once some measurement of quality is accounted for [10].

In our second approach, only using data typically available to decision makers in this setting (NDHS and DHIS2) it was possible to measure three of the recommended steps in the cascade, up to intervention-adjusted coverage. This second approach resulted in lower adjusted coverage estimates at each step of the cascade. Differences in coverage estimates between the two combinations of data sources likely reflect differences in data collection methods and timeframes, with two particular areas of divergence. Regarding inputs, DHIS2 is a census of all facilities and availability of inputs was measured over the last six months, while the project health facility sample survey was conducted at one point in time and reflected availability on the day of survey. Regarding content of care, the second approach did not have the benefit of observations of care which might be considered the most reliable method to capture content of care during childbirth. Rather, it relied on NDHS data on women's reports about care received: this limited the number of items available for the adjustment, plus numerous studies have documented the poor validity of household survey data to assess receipt of interventions [34,45–47].

The results from both approaches highlight that facility readiness to provide care, the second cascade step and a crucial foundation to the provision of high quality of care [6], has not yet been met. Beyond this step the two approaches diverged. While a substantial drop in coverage was estimated from input-adjusted to intervention-adjusted coverage using data typically available in this setting (from 6% to 0.3%) this drop was relatively small in the project data (10% to 7%). And no adjustment for the fourth cascade step, processes of care, was possible using the second approach.

### Strengths and limitations of the data typically available to decision makers

It is not appropriate for countries to routinely generate the comprehensive data that a focussed research project can collect. Nonetheless, there is clearly enormous potential to make better

use of existing data sources for effective coverage measurement. Data on population need and care seeking is readily available from nationally representative population surveys, both DHS and MICS have been widely implemented in LMIC [48,49]. Importantly, these surveys are also designed to be representative at the State level, and as such are frequently used for benchmarking. However, local decision makers often seek more geographical granularity to inform actions; in Gombe state, for example, there is increasing interest to understand variation by LGA to be able to further examine inequalities in access and provision of high quality care across the State and support ongoing quality improvement initiatives. Further, local decision makers prefer more temporal estimates than retrospective household surveys like DHS or MICS can offer, although in this analysis we observed relative stability in access to care in the recent past. To facilitate analysis at lower levels requires alternative sources of population data and strengthening of administration data systems, for example civil registrations and vital statistics and a programme of household surveys to capture information on care seeking [50–52].

It was not possible to measure any components of the processes quality step in the data currently available to decision makers in this setting. Provision of care can be assessed in nationally representative surveys, such as SPA or SARA [26,30]. However, neither currently include observations of childbirth as standard practice, require additional resources to do so, and are susceptible to the issue of temporality [53]. Unlike nationally representative surveys, DHIS2 is a census of all facilities and is available monthly, which offers opportunities to calculate effective coverage measures at the geographical level most useful to decision makers. Data on content of care is not currently present but could potentially be tracked in DHIS2. For example in Gombe State a number of relevant indicators are already captured at the facility level but are not included in the monthly monitoring reports to DHIS2 (see S1 Table) [34]. Extending HMIS so that data beyond inputs is routinely summarised for managers to track depends on government priorities; this may require more advocacy to promote the need for including measures of the content of care.

## Strength and limitations of the approach to measuring effective coverage

The effective coverage cascade is complex and needs further definition. The choice of items included in the effective coverage measure is likely to influence the estimate. Currently there is no standardised list of indicators for measuring quality of maternal and newborn health care [54–56], which poses a challenge to constructing effective coverage as noted by others [15]. Our approach to selecting items to measure each step was highly comprehensive based on a systematic review of the literature, WHO guidelines and cross-checked to ensure relevance to the local context. The measure constructed in the data typically available to decision makers was less comprehensive as not all data items were available (see Table 1), yet the key messages emerging from the analysis were similar.

Once the content of effective coverage measures has been defined (whether comprehensive or pragmatic, according to the data available), the methods for linking datasets for cascade analysis are becoming increasingly clear and accessible. We used validated ecological linking approaches, accounting for facility type, to combine datasets [13,14,42], and variance was estimated using the recommended delta method [43].

## Conclusions

Comprehensive project data revealed that effective coverage of childbirth care in Gombe state is low and more attention is needed on this problem. This study also demonstrates that it is already feasible to partially construct effective coverage measures using routine data from HMIS combined with national level population survey sources. Advocacy to include process

of care indicators in facility summary reports could optimise this data source for local decision making and take us a step closer to operationalising effective coverage measurement at the country level.

## Supporting information

**S1 Table. Components used to define each step of the coverage cascade for the two different data sources.**
(DOCX)

**S1 File. Inclusivity form.**
(DOCX)

## Acknowledgments

The authors would like to thank Data Research and Mapping Consult for coordinating the IDEAS project data collection, the Gombe State Primary Health Care Development Agency (GSPHCDA), Gombe State Ministry of Health and our partners Society for Family Health and Pact Nigeria for their support in carrying out this study. In particularly we would like to thank Christopher Istakis from the GSPHCDA for his support in accessing DHIS2. Finally, we would like to thank all the participants who contributed to our study.

## Author Contributions

**Conceptualization:** Josephine Exley, Tanya Marchant.

**Data curation:** Josephine Exley.

**Formal analysis:** Josephine Exley.

**Methodology:** Josephine Exley, Antoinette Bhattacharya, Claudia Hanson, Abdulrahman Shuaibu, Nasir Umar, Tanya Marchant.

**Writing – original draft:** Josephine Exley, Tanya Marchant.

**Writing – review & editing:** Antoinette Bhattacharya, Claudia Hanson, Abdulrahman Shuaibu, Nasir Umar.

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
