## [Decision Letter · Decision Letter 0]

17 Jan 2022

PGPH-D-21-00996

Operationalising effective coverage of facility based childbirth in Gombe State; a comparison of data sources

Dear Dr. Exley,

Thank you for submitting your manuscript to PLOS Global Public Health. After careful consideration, we feel that it has merit but does not fully meet PLOS Global Public Health’s publication criteria as it currently stands. Therefore, we invite you to submit a revised version of the manuscript that addresses the points raised during the review process.

We look forward to receiving your revised manuscript.

Kind regards,

Victor Adagi Alegana

Academic Editor

Journal Requirements:

1. Please include a complete copy of PLOS’ questionnaire on inclusivity in global research in your revised manuscript. Our policy for research in this area aims to improve transparency in the reporting of research performed outside of researchers’ own country or community. The policy applies to researchers who have travelled to a different country to conduct research, research with Indigenous populations or their lands, and research on cultural artefacts. The questionnaire can also be requested at the journal’s discretion for any other submissions, even if these conditions are not met.  Please find more information on the policy and a link to download a blank copy of the questionnaire here: https://journals.plos.org/plosone/s/best-practices-in-research-reporting. Please upload a completed version of your questionnaire as Supporting Information when you resubmit your manuscript.

2. Please update your Competing Interests statement. If you have no competing interests to declare, please state: “The authors have declared that no competing interests exist.”

3. In the online submission form, you indicated that “The data are available from the principal investigator of the IDEAS project and co-author for this manuscript, Dr Tanya Marchant ORCID id 0000-0002-4228-4334. Reuse permitted on request.”. All PLOS journals now require all data underlying the findings described in their manuscript to be freely available to other researchers, either 1. In a public repository, 2. Within the manuscript itself, or 3. Uploaded as supplementary information.

Additional Editor Comments (if provided):

Kindly respond to reviewers comment regarding data availability. Note PLOS dta policy requires authors to make all data described in maunscript available fully without restriction, with rere exception.

Reviewers' comments:

Reviewer's Responses to Questions

**Comments to the Author**

1. Does this manuscript meet PLOS Global Public Health’s publication criteria? Is the manuscript technically sound, and do the data support the conclusions? The manuscript must describe methodologically and ethically rigorous research with conclusions that are appropriately drawn based on the data presented.

Reviewer #1: Yes

Reviewer #2: Partly

Reviewer #3: Yes

2. Has the statistical analysis been performed appropriately and rigorously?

Reviewer #1: Yes

Reviewer #2: Yes

Reviewer #3: Yes

3. Have the authors made all data underlying the findings in their manuscript fully available (please refer to the Data Availability Statement at the start of the manuscript PDF file)?

Reviewer #1: Yes

Reviewer #2: No

Reviewer #3: Yes

4. Is the manuscript presented in an intelligible fashion and written in standard English?

Reviewer #1: Yes

Reviewer #2: No

Reviewer #3: Yes

5. Review Comments to the Author

Reviewer #1: To the best of what has been presented; the methods and materials used are appropriate to the research objectives. The data analysis techniques used are vigorous and the conclusions have been based on the findings . Therefore, the article meets APLOS publication criteria and is very informative for health practice.

Reviewer #2: Thank you for inviting me to review this manuscript, "Operationalising effective coverage of facility based childbirth in Gombe State; a comparison of data sources". Please find below comments from my review.

(a) Title: The title is not clear. Whereas the study reports findings related to optimising metrics of coverage of maternal health services, this is not accurately represented in the title. I suggest that you include the word(s) "indicator", "metric", "measurement" or whichever you may deem appropriate to clarify that the manuscript content regards metrics and not implementation per se. The term "operationalising" in the title contributes to ambiguity.

(b) Abstract: Whereas the abstract is concise, you need to revise to ensure consistence of tense. Since this is not a study protocol, I suggest using past tense consistently through the manuscript.

(c) Introduction: Lines 54-57. I suggest that you limit this section to clearly stating the study aims. The study design descriptions, such as, ".... using comprehensive facility assessment project data....", can be detailed in the methods section.

(d) Conclusions: Please revise this section to reflect concordance with study aims. Whereas it is true that effective coverage of childbirth care is low in Gombe, this had already been demonstrated from the NDHS, albeit sub-optimally. What then should the reader draw from your work? How did the triangulation of data sources improve (potential) utility of routinely available data?

(e) Study data: please make the study data available or otherwise state justification for not doing so.

Reviewer #3: This study used a combination of facility data to estimate the effective coverage of the childbirth care system in Gombe State, Nigeria. The study found effective coverage estimates for different data combinations and discussed the strength and weaknesses of their method. This adds to the current body of evidence in Gombe State as regards the effective coverage for maternal, newborn, child, and adolescent health and nutrition (MNCAHN). The result is well presented and discussed.

(1.) The authors should consider discussing more on how they managed to adjust for the sample design for multiple data, given that the design could be different for different survey data.

(2.)The authors should explicitly write out the governing formulas/ equations to calculate these measures, and if possible, share the code used in the computation for readers who wish to reproduce the result.

(3.) The authors should proofread again for minor grammatical errors.

The article has sufficient contribution, and I recommend consideration for publication.

6. PLOS authors have the option to publish the peer review history of their article (what does this mean?). If published, this will include your full peer review and any attached files.

**Do you want your identity to be public for this peer review?** For information about this choice, including consent withdrawal, please see our Privacy Policy.

Reviewer #1: No

Reviewer #2: No

Reviewer #3: **Yes: **Bayowa Teniola Babalola

---

## [Editor Report · Decision Letter 1]

22 Mar 2022

Operationalising effective coverage measurement of facility based childbirth in Gombe State; a comparison of data sources

PGPH-D-21-00996R1

Dear Ms Exley,

We are pleased to inform you that your manuscript 'Operationalising effective coverage measurement of facility based childbirth in Gombe State; a comparison of data sources' has been provisionally accepted for publication in PLOS Global Public Health.

Best regards,

Victor Adagi Alegana

Academic Editor